# N-Glycomic and Transcriptomic Changes Associated with CDX1 mRNA Expression in Colorectal Cancer Cell Lines

**DOI:** 10.3390/cells8030273

**Published:** 2019-03-22

**Authors:** Stephanie Holst, Jennifer L. Wilding, Kamila Koprowska, Yoann Rombouts, Manfred Wuhrer

**Affiliations:** 1Center for Proteomics and Metabolomics, Leiden University Medical Center, 2333 ZA Leiden, The Netherlands; 2Cancer and Immunogenetics Laboratory, Weatherall Institute of Molecular Medicine, Department of Oncology, University of Oxford, Oxford OX3 9DS, UK; wildingjl@gmail.com (J.L.W.); kamila.koprowska@gmail.com (K.K.); 3Institut de Pharmacologie et de Biologie Structurale, IPBS, Université de Toulouse, CNRS, UPS, 31077 Toulouse, France; yoann.rombouts@ipbs.fr

**Keywords:** colorectal cancer cell lines, *N*-glycosylation, fucosylation, caudal-related homeobox protein 1 (CDX1), differentiation, fucosyltransferase (FUT), hepatocyte nuclear factor (HNF)4A, HNF1A

## Abstract

The caudal-related homeobox protein 1 (CDX1) is a transcription factor, which is important in the development, differentiation, and homeostasis of the gut. Although the involvement of CDX genes in the regulation of the expression levels of a few glycosyltransferases has been shown, associations between glycosylation phenotypes and CDX1 mRNA expression have hitherto not been well studied. Triggered by our previous study, we here characterized the *N*-glycomic phenotype of 16 colon cancer cell lines, selected for their differential CDX1 mRNA expression levels. We found that high CDX1 mRNA expression associated with a higher degree of multi-fucosylation on *N*-glycans, which is in line with our previous results and was supported by up-regulated gene expression of fucosyltransferases involved in antenna fucosylation. Interestingly, hepatocyte nuclear factors (HNF)4A and HNF1A were, among others, positively associated with high CDX1 mRNA expression and have been previously proven to regulate antenna fucosylation. Besides fucosylation, we found that high CDX1 mRNA expression in cancer cell lines also associated with low levels of sialylation and galactosylation and high levels of bisection on *N*-glycans. Altogether, our data highlight a possible role of CDX1 in altering the *N*-glycosylation of colorectal cancer cells, which is a hallmark of tumor development.

## 1. Introduction

Glycans form an important part of the outer layer of the cell and are involved in major biological processes, including cell differentiation, adhesion, and interactions with other cells, pathogens, or the extracellular matrix, as well as cellular transformations such as cancer [1,2,3]. Glycans occur in many structural variants as modifications on proteins and lipids. One type of glycosylation on proteins is the *N*-glycosylation, in which oligosaccharides are attached via an *N*-glycosidic linkage to an asparagine (N) of the consensus sequence -N-X-S/T- (X is any amino acid except proline, S = serine, T is threonine) [4]. These so-called *N*-glycans share a common pentasaccharide core-structure and form, depending on the elongation, high-mannose type, complex-type, or hybrid-type *N*-glycans (Figure 1A) [4]. Changes of specific *N*-glycans and other glycans have been associated with several diseases and cancers [5,6,7] and colorectal cancer associated glycan changes have been recently reviewed by us [8]. 

By screening different colorectal cancer cell lines, we previously showed an association of caudal-related homeobox protein 1 (CDX1) mRNA expression with increased multi-fucosylation (more than one fucose), which is indicative for antenna fucosylation [9]. Antenna fucosylation is the result of the activity of fucosyltransferases encoded by the genes *FUT1, 2, 3, 4, 5, 6, 7*, and *9* and leads to the expression of blood group related Lewis antigens (Figure 1B), which are glycan-epitopes on various glycoproteins and glycolipids [10,11]. While *FUT3,4,5,6,7,* and *9* catalyze the addition of a fucose to the *N*-acetylglucosamine (GlcNAc) of the antenna in α1,3- and/or α1,4-linkage, *FUT1* and *FUT2* are responsible for the addition of a fucose to the galactose (Gal) in α1,2-linkage, forming the H-type epitope. *FUT2* is further called the secretor gene and polymorphisms leading to an inactive *FUT2* lead to the absence of blood type epitopes in saliva and various epithelial cell types, the so-called non-secretor phenotype [12,13]. *FUT8*, on the other hand, is the gene encoding for the enzyme mediating the attachment of a fucose-residue to the first GlcNAc of the *N*-glycan core, the so-called core-fucosylation. More detailed information on the biological role of fucosylation has been given elsewhere [14].

Fucosyltransferases are expressed in a cell type- and organ-dependent manner and altered expression of the enzymes, as well as the produced fucose-containing glycan epitopes, have been associated with various pathological conditions [15], including colorectal cancer [16,17]. However, literature remains controversial concerning the stage-dependent differences in expression of fucosylated glyco-epitopes in colorectal cancer. Conflicting data arises from various experimental conditions (lectin and antibody based vs. mass spectrometric detection, sample preparation, and culture conditions) and sample sources (cell lines, mouse models, tissues vs. serum or plasma, biopsy location) as well as combining results from studies on various glycan classes (*N*-, *O*-, and glycosphingolipid glycans) vs. individual glycan classes. Moreover, core- and antenna-fucosylation are not always distinguished and only a few studies report on stage-dependent changes. While many groups reported a general increase of fucosylation in colon and other cancers [15], other recent studies have shown a stage-dependent decrease of fucosylation, especially Lewis-type antenna-fucosylation, with colon cancer metastasis [18,19]. Notably, it is still largely unclear to which extent differences in the glycosylation machinery, on the one hand, and differences in the expression levels of highly glycosylated proteins such as mucins (e.g., MUC2), on the other hand, contribute to these colorectal cancer glycosylation phenotypes.

The group around Miyoshi and Moriwaki hypothesize that deficiency of fucosylation via mutations of GDP-mannose-4,6-dehydratase (*GMDS*), an important player in the fucose biosynthesis pathway, helps cancer cells to escape from natural killer (NK) cell-mediated tumor immune surveillance [20,21]. Interestingly, mutations in the *GMDS* gene have been identified in metastatic lesions of some colon cancers (10%). Also the colon cancer cell line HCT116 bears a *GMDS* mutation, resulting in almost complete absence of fucosylation. Strikingly, the parental HCT116 cells with *GMDS* mutation revealed a more aggressive phenotype in tumor formation and metastasis in mice, as compared to the *GMDS*-rescued HCT116 cells [17]. The group speculated that the loss of fucosylation leads to an escape from NK cell-mediated tumor immune surveillance, while the *GMDS*-rescued HCT116 were more susceptible to TRAIL-induced apoptosis [21].

For the sialylated variants of the Lewis antigens (Figure 1B), sialyl Lewis X and sialyl Lewis A, the literature is clearer. The selectin ligands sialyl Lewis X and A have been associated with advanced stages and poor prognosis in colon cancer [22,23] which were attributed to selectin-mediated adhesion and extravasation leading to metastasis. Moreover, sialyl Lewis antigens are used as clinical markers for various cancers (e.g., sialyl Lewis A = CA19-9) [16].

CDX1 is a transcription factor that is involved in the modulation of a variety of processes including proliferation, apoptosis, cell-adhesion, and columnar morphology [24]. CDX1 is a primary controller of enterocyte differentiation and its expression is necessary for the transcriptional regulation of a large number of intestine-specific genes [25] required for the intestine development, differentiation, and maintenance of the intestinal phenotype [24,26]. Several markers of differentiation, including villin and cytokeratin 20, have been shown to be directly transcriptionally regulated by CDX1 [27,28]. Loss of differentiation is known to occur during cancer progression and, in colon cancer, the loss of villin has been associated with poor survival [29].

There is evidence of the loss or down-regulation of CDX1 expression in colon cancer tumors [30,31] and cell lines [32]. With respect to the underlying mechanism, Wong et al. have shown that the loss or reduction of CDX1 is often induced by promoter methylation [33]. Together, these observations indicate a potential role of CDX1 loss in tumor development.

Although glycosylation as well as CDX1 expression have both been shown to play a role in cell differentiation and cancer (suppression), hitherto only very little has been described with regard to CDX1-associated glycosylation profiles. Triggered by our previous findings [9], which suggested a relationship between increased multi-fucosylation (Lewis type glycan epitopes) and high CDX1 mRNA expression, here we characterized the *N*-glycosylation phenotype of a different set of colon cancer cell lines. These cells were specifically chosen for their high and low expression of CDX1 mRNA, with the aim of replicating our previously found association of fucosylation and CDX1 expression [9] with an independent set of cell lines. The glyco-phenotypic data were further correlated with differently expressed glyco-genes. 

We confirmed a higher degree of multi-fucosylation in cell lines with high CDX1 gene expression, which was supported by higher mRNA expression of fucosyltransferases *FUT3* and *FUT6*, both involved in antenna fucosylation. Strikingly, other genes associated with fucosylation were likewise differentially expressed between the investigated cell lines with high vs. low CDX1 expression. A higher *GMDS* gene expression is positively associated with high CDX1 mRNA expression in the tested colorectal cancer cell lines and also transcription factors hepatocyte nuclear factor (*HNF*)*1A* and *HNF4A*, which have been previously shown to regulate antenna fucosylation of human plasma proteins [34] and are associated with differentiation [35,36], showed higher expression in CDX1-high expressing cells. Furthermore, the *N*-glycomic characterization revealed a decrease in galactosylation and sialylation in cell lines with high CDX1 expression, compared to cells with low CDX1 gene expression.

## 2. Materials and Methods

### 2.1. Materials

8M guanidine hydrochloride (GuHCl), 1-hydroxybenzotriazole (HOBt) hydrate, 50% sodium hydroxide (NaOH), and super DHB matrix (2-hydroxy-5-methoxy-benzoic acid and 2,5-Dihydroxybenzoic acid, 1:9) were obtained from Sigma-Aldrich (Steinheim, Germany). HPLC SupraGradient acetonitrile (ACN) was obtained from Biosolve (Valkenswaard, The Netherlands). Dithiothreitol (DTT), ethanol, and sodium bicarbonate (NaHCO_3_) were from Merck (Darmstadt, Germany) and 1-ethyl-3-(3-dimethylaminopropyl) carbodiimide (EDC) was from Fluorochem (Hadfield, UK). Peptide *N*-Glycosidase F (PNGase F) was purchased from Roche Diagnostics (Mannheim, Germany). The peptide calibration standard was purchased from Bruker Daltonics (Bremen, Germany). MultiScreen^®^ HTS 96 multiwell plates (pore size 0.45 μm) with high protein-binding membrane (hydrophobic Immobilon-P PVDF membrane) were purchased from Millipore (Amsterdam, The Netherlands), while the 96-well polypropylene 0.8 mL 96-deepwell plate and 96-well PCR plate polypropylene were from Greiner Bio (Alphen a/d Rijn, The Netherlands). All buffers were prepared using ultrapure water generated from an 18.2 MΩ-cm Purelab Ultra system from Elga (Ede, The Netherlands). Control Visucon-F plasma pool (citrated and 0.02 M HEPES buffered plasma pool from 20 healthy human donors) was obtained from Affinity Biologicals (Ancaster, Canada). Cell culture media were purchased from Gibco (Paisley, UK), fetal bovine serum (FBS) from Autogen Bioclear (Wiltshire, UK), and penicillin-streptomycin from Invitrogen (Paisly, UK). The RNeasy mini kit was purchased from Qiagen (Manchester, UK).

### 2.2. Cells and Cell Culture

Details of colorectal cancer cell line origin and characteristics can be found in Table 1 and more detailed in Appendix A. CC20, COLO678, GP2D, HCT116, ISRECO1, LIM1863, LS174T, PCJW, RCM1, and SW403 cell lines were cultured in the Dulbecco Modified Eagle Medium (DMEM). CAR1, HCA46, HDC8, OXCO1, and VACO429 were cultured in Iscove’s Modified Dulbecco’s Medium (IMDM). HCC56 cells were cultured in the RPMI-1640 medium. All of the above media were supplemented with 10% FBS and 1% penicillin-streptomycin. Cell lines were grown in a 10% CO_2_ incubator to 50% to 80% confluence before next passage or cell pellet preparation. Cell pellets were washed once with PBS, snap-frozen in dry ice, and stored at −80 °C until further analysis. Cell pellets were dissolved in water and cell counts were estimated using the Countess^TM^ Automated Cell Counter (Invitrogen), based on trypan blue staining.

### 2.3. N-glycan Release, Derivatization, and Purification

*N*-glycans were released in duplicate from three biological replicates per cell line using a PVDF-membrane based release protocol followed by linkage-specific sialic acid derivatization. Purification by cotton-HILIC-SPE and MALDI-TOF-MS analysis was performed as described previously [9]. Shortly, cell pellets were resuspended in water, disrupted by sonication, and proteins immobilized on 96-well PVDF-filter plates (~0.5 × 10E6 cells/well; 5µL human plasma control; water as blanks) in the presence of chaotropic agents. After washing unbound materials, *N*-glycans were released overnight at 37 °C. Released glycans were derivatized using ethyl esterification allowing for discrimination of *N*-acetylneuraminic acid linkages (α2,3 vs. α2,6) [37] in a ratio of 20:100 (sample: derivatization reagent), and purified by cotton-thread HILIC-SPE [37].

### 2.4. MALDI-TOF-MS Analysis

Released, derivatized, and purified glycans (5 µL sample) were spotted on an anchor chip MALDI target plate (Bruker Daltonics) and co-crystallized with 0.5 µL of 5 mg/mL superDHB in 50% can, supplemented with 1 mM NaOH. Spectra were recorded in positive-ion reflector mode, after calibration with a Bruker peptide calibration kit, using a Bruker UltrafleXtremeTM mass spectrometer, controlled by FlexControl 3.4 software Build 119 (Bruker Daltonics). Mass spectra were obtained over an *m/z* range of 1000 to 5000 for a total of 10 000 shots (1000 Hz laser frequency, 200 shots per raster spot during complete random walk). Tandem mass spectrometry (MALDI-TOF-MS/MS) was performed for structural elucidation via fragmentation in gas-off TOF/TOF mode.

### 2.5. Data Processing and Analysis of MALDI-TOF-MS Spectra

Spectra were smoothed (Savitzky Golay algorithm, peak width: *m/z* 0.06, 4 cycles), baseline corrected (Tophat algorithm), and exported to xy-files using FlexAnalysis 3.4 (Stable Build 76). Mean average spectra were generated per technical replicate, which were summed to one spectrum using the open-source software mMass (http://www.mmass.org; [38]). The spectrum was internally re-calibrated using glycan peaks of known composition (H5N2 at *m/z* 1257.423, H6N2 at *m/z* 1419.476, H7N2 at *m/z* 1581.529, H8N2 at *m/z* 1743.581, H5N4F1 at *m/z* 1809.639, H5N4F2 at *m/z* 1955.697, H5N4E1 at *m/z* 1982.709, H10N2 at *m/z* 2067.687, H6N5F1 at *m/z* 2174.7715, H5N4L1E1 at *m/z* 2255.793, H5N4E2 at *m/z* 2301.835, H6N5E1 at *m/z* 2347.8403, H7N6F1 at *m/z* 2539.904, H6N5F4 at *m/z* 2612.945, H6N5L1E1 at *m/z* 2620.925, H7N6E1 at *m/z* 2712.973, H7N6L2 at *m/z* 2940.016, H9N8 at *m/z* 3124.111, H7N6L4F1 at *m/z* 3632.243) as calibrants (minimum five used), followed by peak picking in mMass, with cut-off signal-to-noise (S/N) 3. The peaklist was manually revised and analyzed in GlycoWorkbench 2.1 stable build 146 (http://www.eurocarbdb.org/) using the Glyco-Peakfinder tool (http://www.eurocarbdb.org/ms-tools/) for generation of a glycan compositions list. Our novel in-house software, developed for automated data processing, MassyTools version 0.1.8.0 [39], was used for calibration using a 3^rd^ degree function and targeted data extraction of the area under the curve for each individual mass spectrum. To prevent over-estimation of overlapping glycan species, only the first three isotopes were extracted and the area under the curve was corrected based on the theoretical isotopic pattern. The quality of the data was assessed using several quality parameters calculated within the software. Only good quality spectra (total intensity > 1 × 10^5^ and fraction of analyte area with S/N > 9 is more than 50%) as well as analytes (S/N > 6, ppm < 20, quality score > 0.10) were included for analyses. Raw data after pre-processing is provided in the Supplementary Tables. Finally, the corrected area-under-the-curve values were rescaled to a total relative intensity of 100% for each spectrum. Selected glycan compositions were confirmed by MS/MS and a final peak list as well as MS/MS data is given in Appendix A. MS/MS spectra were manually interpreted and fragment ions annotated using GlycoWorkbench 2.1 according to the nomenclature of Domon and Costello [40]. Averages of direct traits per cell line were used to build a principle component analysis model in SIMCA Version 13.0 (Umetrics AB, Umea, Sweden), with seven random cross-validation (CV) groups.

For increased robustness, derived glycan traits such as galactosylation, fucosylation, sialylation, and others were calculated in SPSS Version 23 (IBM Corp, Armonk, NY). The formulas for calculation are given in Appendix A and the average relative abundances are given in Appendix A. Due to non-normally distributed data, a two-tailed Mann–Whitney test was performed in Rstudio statistical software environment (Version 0.99.892, Kent, OH, USA, http://www.r-project.org/) with the significance level α = 0.05 to assess differences in *N*-glycosylation between CDX1 high and low expressing cells. Bonferroni correction was applied to *p*-values to adjust for multiple testing (Appendix A). Boxplots for visualization were generated in Rstudio and show the median with the interquartile range.

### 2.6. Gene Expression Microarrays and Data Analysis

Total RNA was extracted by using the RNeasy mini kit according to the manufacturer’s instructions. Twenty micrograms of RNA of each sample were sent to the Molecular Biology Core Facility of the Paterson Institute for Cancer Research, Manchester, UK, for gene expression microarray analysis using the Human genome U133+2 chips, following the manufacturer’s instructions (Affymetrix, High Wycombe, UK). Microarray data were analyzed using Partek Genomics Suite software. The data were log2-transformed and RMA-normalized (with GC correction) using quantile normalization with Median Polish for Probeset summarization as optional settings in the software. 

Glycosyltransferase genes (Appendix A) as well as glycan-related genes (Appendix A) were selected for analysis and differentially expressed genes were identified from a *t-test* comparing mean-expression levels of the 8 CDX1^high^ vs. 8 CDX1^low^ cell lines. The significant level was adjusted for multiple testing. Fold changes were calculated for CDX1^high^ and CDX1^low^ cell lines and for significantly different expressed glycosyltransferases, data was visualized as boxplots in Rstudio showing the median and interquartile range. To evaluate the correlation between relative abundances of *N*-glycans traits based on mass spectrometry data and glycosyltransferase gene expression in the 16 investigated cell lines, linear regression analysis was performed in GraphPad Prism Version 6 (GraphPad Software, Inc., La Jolla, CA). 

## 3. Results

### 3.1. CDX1high and CDX1low CRC Cells Exhibit Different N-Glycan Profiles 

Our previous data suggested a positive association between the level of fucosylation on protein *N*-glycans and CDX1 mRNA expression in a set of colorectal cancer cell lines (15 cell lines CDX1/villin positive, 7 cell lines CDX1/villin low or negative) [9]. To validate our previous results and to further explore expression profiles of associated fucosyltransfereases, we characterized the *N*-glycosylation of an independent set of colorectal cancer cell lines with high (8 cell lines; CDX1^high^) vs. low (8 cell lines; CDX1^low^) expression of CDX1 mRNA. The CDX1^high^ cell lines investigated here have, on average, 65-fold higher CDX1 mRNA expression as compared to CDX1^low^ cell lines (Table 1, Appendix A). Two exemplary mass spectra of the CDX1^low^ cell line Colo678 and the CDX1^high^ cell line HCA46 are shown in Figure 2. As observed in other cell line profiling studies, the *N*-glycomic profiles of all analyzed cell lines were dominated by high-mannose type *N*-glycans, but differed in the ratios of these glycans. With regard to complex type *N*-glycans, those derived from Colo678 exhibited more sialic acid residues (*N*-acetylneuraminic acid, NeuAc, purple diamond, angle indicates different linkages), especially in α2,3-linkage, while glycans from HCA46 were characterized by the presence of many fucoses (Fuc, red triangle), low sialylation level, and additional *N*-acetylhexosamines (HexNAc, white square). In total, 221 individual glycan species were identified across all cell lines, from which 81 could be characterized by tandem mass spectrometry (Appendix A). In order to evaluate whether CDX1^high^ and CDX1^low^ expressing cells can be distinguished based on their *N*-glycomic signature, a principal component analysis (PCA) was performed on the relative abundances of all individual *N*-glycans. This resulted in a model with four principle components (PC) explaining 68.9% of the variation in the data. The score plot of PC1 vs. PC2 showed clear separation of CDX1^high^ and CDX1^low^ cell lines along PC1 (Figure 3A) in this unsupervised model, demonstrating a different *N*-glycan profile between the two groups.

### 3.2. Higher Antenna Fucosylation on N-Glycans Characterized CDX1-high Expressing Colorectal Cancer Cell Lines 

To assess which glycans drive the principal component separation of CDX1^high^ and CDX1^low^ colorectal cancer cells, derived glycan traits were calculated by grouping glycan species (direct traits) into classes according to glycosylation characteristics, such as sialylation and fucosylation. The relative abundances of derived traits as well as the calculations are given in Appendix A. Derived traits gave insight into general structural differences and showed increased analytical robustness as compared to individual glycans. Observations from the example spectra were confirmed by comparing derived glycan traits of CDX1^high^ and CDX1^low^ expressing cell lines and evaluated using a Mann–Whitney test. The *p-values* were adjusted for multiple testing (Bonferroni). Labelling of the PC1 vs. PC2 score plots using glycan-derived traits showed that multi-fucosylated glycans (Figure 3B, colored in green) associated with the location of CDX1^high^ cell lines in the score plot, in line with observations from the two exemplary mass spectra (Figure 2). Accordingly, CDX1^high^ expressing cell lines exhibit significantly higher levels of multi-fucosylation (presence of more than one fucose on a glycan), indicative for antenna fucosylation (Lewis X/A, Y/B), as compared to CDX1^low^ expressing cell lines (∅ 54% vs. ∅ 33%; *p-value* = 0.011; Figure 4A, Appendix A) and confirmed the association found in our previous study [9]. In order to map the *N*-glycomic phenotype to transcriptomic data, a gene microarray was performed and glycosyltransferase gene expression data was extracted for the 16 investigated cell lines (Appendix A). Further, a linear regression analysis was used to evaluate the correlation between MS-based *N*-glycan traits and glycosyltransferase gene expression data (Appendix A). The trait CFa, representing multi-fucosylation in complex type *N*-glycans, showed significant correlation with fucosyltransferase genes *FUT2, 3, 4, 6,* and *7* which are involved in antenna fucosylation (Appendix A), thereby indicating that the trait CFa is a good representation for antenna fucosylation in this data. The glycosyltransferase gene expression was also tested for differential expression between CDX1^high^ and CDX1^low^ cell lines and, in accordance with mass spectrometry data, all fucosyltransferases involved in antenna-fucosylation showed higher expression in CDX1^high^ cell lines, though after correction for multiple testing only *FUT3* and *6,* involved in Lewis X/A biosynthesis, showed significantly increased expression (2.7- to 5.7-fold) in the eight CDX1^high^ cell lines, compared to eight CDX1^low^ cells (Figure 5A,B, Appendix A). In contrast, mono-fucosylation, indicative for core-fucosylation (CFc), was lower in CDX1^high^ cells as compared to CDX1^low^ cells and also FUT8 gene expression showed a trend towards lower expression in CDX1^high^ cells (Appendix A). 

Furthermore, the GDP-L-fucose precursor GDP-mannose 4,6-dehydratase (*GMDS* gene) gene expression was 3.7-fold (*p-value* 1.4 × 10E-03) elevated with high CDX1 expression (Figure 5C; Appendix A). The corresponding enzyme GDP-mannose 4,6-dehydratase is involved in the fucosylation process, indicating that various enzymes involved in fucosylation are differentially regulated in CDX1^high^ versus CDX1^low^ cells. As the transcription factors *HNF1A* and *HNF4A* have previously been shown to regulate antenna fucosylation in plasma [34] and have previously been associated with CDX1 expression as well as intestinal development [41,42,43], here we tested the expression levels in the investigated CDX1^high^ and CDX1^low^ cell lines. Interestingly, gene microarray data revealed significantly increased expression of *HNF1A* and *HNF4A* with high CDX1 mRNA levels (*p-value* (HNF1A) = 0.002; *p-value* (HNF4A) = 0.003; Figure 5J–K, Appendix A). Moreover, soluble galectin 4 (*LGALS4*), a target gene of *HNF4A* and a glycan-binding protein, was 23-fold up-regulated in CDX1^high^ cells (*p-value* = 2.4 × 10E-06; Figure 5L, Appendix A). Galectin 4 is highly expressed in the alimentary tract during the development and is associated with differentiation [44].

### 3.3. CDX1high Expressing CRC Cell Lines Exhibit a Lower Level of Sialylated N-Glycans as Compared to CDX1-Low Expressing Cells

In our previous study, we observed a trend towards a negative association between overall *N*-glycan sialylation and CDX1 mRNA expression, though with no significant difference after correction for multiple testing. To further explore this association, we labelled the score plot of PC1 vs. PC2 with derived sialylation traits and found that sialylated glycans largely marked CDX1^low^ cell lines (Figure 3C, colored in blue and red). In agreement, overall sialylation was significantly lower in CDX1^high^ cell lines compared to CDX1^low^ expressing cell lines (∅ 36% vs. ∅ 21%; *p-value* = 0.046; Figure 4B, Appendix A). Results from the current study showed pronounced sialylation differences for α2,3-sialylated *N*-glycans with ∅ 11% in CDX1^high^ vs. ∅ 23% in CDX1^low^ cell lines (*p-value* = 0.023; Figure 4C; Appendix A). Different sialyltransferases are involved in this sialylation and mRNA levels of sialyltransferases *ST3GAL3* and *6* correlated with relative abundances of overall sialylation in complex type *N*-glycans (Appendix A). In line with results from MS data, *ST3GAL3,4* and, especially, *ST3GAL6,* all three involved in sialyl Lewis antigen biosynthesis, were decreased in CDX1^high^ cell lines as compared to CDX1^low^ cell lines, though not significantly (Figure 5I, Appendix A). Notably, sialylation per galactose in the MS data was likewise decreased from ∅ 48% (CDX1^low^) to ∅ 31% in CDX1^high^ cell lines (*p-value = 0.023*; Appendix A), suggesting the decrease in sialylation being an independent event and not a mere result of substrate limitation through decreased galactosylation (see below).

### 3.4. CDX1 Expression in CRC Cell Lines Associated with N-Glycans CARRYING Additional N-Acetylhexosamine 

While differences in levels of fucosylation and sialylation mainly explain the separation of CDX1^high^ and CDX1^low^ CRC cells along PC1, we sought to determine whether other glycan-derived traits could account for the separation in other PC dimensions. Glycan structures with the number of HexNAc equaling or exceeding the number of hexoses (Hex; HexNAc≥Hex) represent glycans with bisecting *N*-acetylglucosamine (GlcNAc), non-galactosylated antennae, or the addition of *N*-acetylgalactosamine (GalNAc) and showed a significantly increased expression in CDX1-positive cells in our previous study [9]. This could be validated in the new set of cells, in which CDX1^high^ cell lines showed a more than 2-fold increase in the relative abundance of glycans with the feature HexNAc≥Hex, compared to CDX1^low^ cells (*p-value* = 0.023; Figure 4D; Appendix A). Corresponding gene expression of the glycosyltransferase involved in the formation of bisecting GlcNAc (*MGAT3*) was significantly correlated with MS data (Appendix A), suggesting the presence of bisection and refining the data obtained by mass spectrometry. However, *MGAT3* gene expression showed only a trend towards increased expression in CDX1^high^ cells compared to CDX1^low^ cells (Figure 5D, Appendix A).

Other glycan epitopes potentially contributing to the HexNAc≥Hex class are the LacdiNAc (GalNAcβ1–4GlcNAc) epitope, the SdA antigen (NeuAcα2–3(GalNAcβ1–4)Galβ1–4GlcNAc; NeuAc=*N*-acetylneuraminic acid, Gal=galactose), and the blood group A epitope (GalNAcα1–3(Fucα1–2)Galβ1–3/4GlcNAc). Accordingly, one of the glycosyltransferases adding a GalNAc-residue to a GlcNAc to form LacdiNAc structures on *N*-glycans, *B4GALNT3*, was significantly correlated with the MS-based trait HexNAc≥Hex (Appendix A) and was, additionally, significantly higher expressed in CDX1^high^ versus CDX1^low^ cells (Figure 5E, Appendix A). Other genes involved in the expression of these epitopes were not significantly different (Appendix A).

While glycans with additional HexNAcs where elevated, galactosylation per antenna was found to be significantly lower in CDX1^high^ cells than in CDX1^low^ cell lines (∅ 85% vs. ∅ 72%; *p-value* = 0.046; Figure 4E, Appendix A). Corresponding galactosyltransferases showed a corresponding trend (Appendix A). 

### 3.5. CDX1 Expression in CRC Cell Lines Associated with Higher Branched N-Glycan-Derived Traits 

Finally, *N*-glycan structures with seven or more HexNAcs, indicative for branched structures or (poly-) LacNAc repeats (–Galβ1–4GlcNAcβ1–3Galβ1–3/4–GlcNAc–), showed a trend towards higher expression with high CDX1 expression in our previous data and were higher expressed in CDX1^high^ cell lines of the new set of cells as compared to CDX1^low^ expressing cell lines, with ∅ 27% vs. ∅ 20% (Figure 4F, Appendix A), though not significantly after multiple testing correction. Since our MS data could not sufficiently differentiate between LacNAc-repeat and additional antenna, we next analyzed the glycosyltransferase expression data to see if this would give a more detailed insight. Genes encoding for beta-1,3-*N*-acetylglucosaminyltransferase 3 (*B3GNT3*) and *B3GNT8*, both involved in the synthesis of type-1 chains and poly-LacNAc repeats, were around 2-fold up-regulated with high CDX1 expression (Figure 5F+G, Appendix A) and *B3GNT3* gene expression showed significant correlation with the MS trait HexNAc≥7 (Appendix A), suggesting the presence of LacNAc-repeat structures, to a certain degree. Additionally, the glycosyltransferase encoded by gene *MGAT4A*, which is involved in the branching on the 1,3-arm of *N*-glycans to form tri- and tetra-antennary *N*-glycans, was correlated with the relative abundance of the MS *N*-glycan trait HexNAc≥7 (Appendix A). *MGAT4A* was further significantly higher expressed in CDX1^high^ cells as compared to CDX1^low^ cells (Figure 5H, Appendix A). Of note, *N*-glycans featuring HexNAc≥Hex also contribute to this trait of *N*-glycan structures with seven or more HexNAcs, which is therefore not solely indicative for branching and poly-LacNAc repeats. In line, *B3GNT3* and *B3GNT8* also showed correlation with the MS trait HexNAc≥Hex (Appendix A).

Overall, we could validate our previous results and identify specific *N*-glycan features associated with high CDX1 mRNA expression, which were characterized by high multi-fucosylation, elevated levels of *N*-glycans containing additional HexNAcs as well as low galactosylation and low sialylation levels, with particularly decreased levels of α2,3-sialylation. 

## 4. Discussion

In two independent sets of colorectal cancer cell lines we observed increased multi-fucosylation in CDX1^high^ expressing cell lines, next to increased terminal HexNAc epitopes as well as decreased galactosylation and sialylation. Changes in glycosylation have mainly been attributed to alterations in the corresponding glycosyltransferases, as also described in literature [45,46]. Although several glycosyltransferase gene expressions correlated well in the study presented here, predictions on the glycan phenotype using gene expression data remain challenging since several biological effectors can influence not only the expression of glycan-initiating, -elongating, and -degrading enzymes, but also the enzyme activity [47]. 

Guo and Pierce, for example, recently reviewed the involvement of transcription factors in glycan expression and reported that they regulate the expression of glycosyltransferase genes in a tissue- and cell-specific manner [47]. Interestingly, we found CDX1, a colon-specific transcription factor involved in cell differentiation, associated with different glycosylation features. Cell lines expressing high mRNA levels of CDX1 showed an *N*-glycan phenotype with increased multi-fucosylation, indicative for antenna-fucosylation. Supporting this association, fucosyltransferase genes *FUT3* and *6*, and *GMDS*, which is involved in GDP-L-fucose synthesis, were positively associated with high CDX1 expression. Differences were more pronounced for *FUT3*, suggesting an enhanced expression of the type-1 chain epitope Lewis A in CDX1^high^ cells [48]. 

Our data suggests antenna fucosylation on *N*-glycans as a potential marker for epithelial differentiation in colorectal cancer cell lines. In line with our results here, a previous study using HCT116, which is characterized by a poor differentiation status and has a mutation in the *GMDS* gene, leading to low levels of fucosylation, showed that restoration of *GMDS* wild type expression enhanced fucosylation, suppressed tumor formation and reduced the metastatic potential, when injected into mice [20]. Additionally, HCT116 cells have shown higher levels of cancer stem cells (CSC) and loss of CDX1 expression, whereas induced expression of CDX1 led to reduced clonogenicity and restored the potential of cells for differentiation and lumen formation [49]. Supporting the hypothesis of fucosylation as a characteristic of epithelial cells, Breiman et al. identified fucosylated antigens in the mammary cancer cell line as markers of the epithelial state, which can contribute to cell adhesion through CLEC17A (Prolectin) [50]. During the epithelial-to-mesenchymal transition (EMT) of these cells, the expression of fucosylated antigens such as Lewis Y was decreased as a result of decreased expression of fucosyltransferases encoded by *FUT1* and *FUT3* genes [50]. 

The transcription factors *HNF1A* and *HNF4A* were also positively associated with high CDX1 mRNA expression in the examined cell lines. Lauc et al. identified in a genome-wide association study (GWAS) that fucosyltransferases *FUT3, 5*, and *6,* all involved in antenna fucosylation, are positively regulated by *HNF1A* and its downstream factor *HNF4A*, while *FUT8*, initiating core-fucosylation, is inhibited [34]. The study showed that knock-down of *HNF1A* and *HNF4A* resulted in reduced expression of several fucosyltransferase genes involved in antenna fucosylation in HepG2 liver cells and, partly, in pancreatic Panc1 cells, while *FUT8*, responsible for *N*-glycan core-fucosylation, was upregulated. Furthermore, *GMDS* and *L-Fucokinase,* both involved in two different pathways of fucose synthesis, were drastically down-regulated upon *HNF1A* or *HNF4A* knock-down. Publicly available ChIP-seq data of ENCODE show that *HNF4A* and *HNF4G* transcription factors bind to *FUT2, 3*, and *6* genes in cell lines of non-intestinal origin and, also, Lauc et al. confirmed the binding of *HNF1A* and *HNF4A* to multiple *FUT* genes and/or their promotors by ChiP analysis [34]. At the same time, several groups reported on the interaction between CDX1 as well as the highly related, partially redundant, CDX2 with other transcription factors, including *HNF1A* and *HNF4A*. Boyd et al. identified CDX2 binding sites on CDX1 and *HNF1B*, both being positively regulated by CDX2, while *HNF1B* is needed for activation of *HNF4A, HNF1A,* and *HNF3G* [43]. Direct binding of CDX2 to *HNF4A* was also shown to positively influence transcriptional activity [43]. *HNF4A* was identified as a transcriptional activator for intestinal differentiation and gene expression of *HNF4A* and its target genes were upregulated in colorectal cancer cell lines with a more epithelial phenotype, as compared to cells with a mesenchymal phenotype [51]. One target gene of *HNF4A* is galectin 4, which was described as tumor-suppressor in colon cancer [52,53], but also pancreatic duct adenocarcinoma [54], and was accordingly positively associated with high CDX1 mRNA expression in the current study. 

Taking our findings and the reports from literature, one may speculate that antenna fucosylation of *N*-glycans is a marker for the epithelial state and that CDX1 is involved in the transcriptional regulation of fucosyltransferases involved in (antenna-)fucosylation, leading to a more differentiated and less invasive phenotype in colorectal cancer cell lines. 

Interestingly, in our investigated cell lines, CDX2 mRNA expression correlated with CDX1 mRNA expression (both high or both low), with the exception of the cell line RCM1. Consequently, similar observations as for CDX1-associated glycosylation were made for CDX2, but correlations with the *N*-glycan phenotype and glycosyltransferase expressions were less pronounced for CDX2 as compared to CDX1 (data not shown). In contrast, a direct involvement of CDX2 in the regulation of *FUT2*, which revealed potential binding sites for CDX1 and CDX2 and is involved in the generation of LeY/B antigens, was shown in colon cancer cell lines HT29 and DLD-1 [55]. Expression of CDX2, and thereby *FUT2*, could be reduced through treatment with epidermal growth factor (EGF)/bFGF [55]. On the other hand, the sialyl Lewis antigen promoting glycosyltransferases *ST3GAL1/3/4* and *FUT3* were transcriptionally up-regulated by c-Myc [55]. While sialyl Lewis types antigens are commonly associated with (colorectal) cancers [8], the CDX1^high^ cell lines show a very low expression of α2,3-sialylation, thereby limiting the substrate for specific fucosyltransferases involved in the expression of sialyl Lewis epitopes (*FUT3,5,6,7).* The combined expression of α2,3-sialylation and antenna fucosylation, as reflected in the trait CLFa, may be considered a proxy for sialyl Lewis epitope expression. Both groups of cell lines in the present study only showed low levels of CLFa (~4%), indicating low levels of sialyl Lewis epitopes. In the case of CDX1^low^ expressing cell lines, the combination of α2,3-sialylation with overall fucosylation increased (minimum of one fucose which may be core or antenna attached; ∅ 16%), which may likewise reflect increased sialyl Lewis epitope levels. Strikingly, in CDX1^high^ expressing cell lines the expression of multi-fucosylation (indicative for antenna fucosylation) and α2,3-sialylation of *N*-glycans even showed opposite trends. Though *FUT6* seems to have a preference for the sialylated substrate, here, the results point towards the expression of mainly non-sialylated Lewis antigens via action of fucosylytransferases *FUT3* (Lewis A) and *FUT4* or *FUT6* (Lewis X) [48] in the CDX1^high^ colorectal cancer cell lines analyzed here.

Especially in the context of sialyl Lewis epitopes, the importance to distinguish α2,3- and α2,6-sialylation becomes evident. The derivatization applied here differentially modifies the sialic acids in different linkages resulting in a detectable mass shift allowing the distinction between α2,3- and α2,6-sialylation [37]. Similar to α2,3-linked sialic acid, α2,8-linked sialic acid will be lactonized under the conditions of the derivatization step [56]. We therefore cannot differentiate between α2,3-linked sialic acid and α2,8-linked sialic on the basis of the observed masses alone. However, we performed in-depth MS/MS characterization of *N*-glycans from different colorectal cancer cells and could not find evidence for a fragment corresponding to an antenna with two or more sialic acids [9]. Although *N*-glycans with terminal α2,8-linked sialic acids are expressed in cancer cells, these structures might be expressed at low levels as compared to *N*-glycans with terminal α2,6- and α2,3-linked sialic acids. Moreover, gene expression of ST8SIA1-5 was not significantly different between the here studied CDX1 high and CDX1 low expressing colorectal cell lines. Sialic acids can further be modified by O-acetyl-groups modulating the ligand function. O-acetylated glycans have been reported to be mainly present in the lower part of the intestinal tract [57]. Furthermore, a decrease of O-acetylation has been observed with colon cancer progression [58]. In line, we previously detected O-acetylated sialic acids of glycosphingolipid glycans, which were decreased in colorectal cancer tissues as compared to control tissues [59], but not on *N*-glycans of the same tissues [1]. The derivatization method applied in this study to characterize the *N*-glycans of colorectal cancer cell lines has been shown to preserve O-acetylation [60]. However, we could not find evidence of O-acetylated *N*-glycans in the investigated cell lines.

We further observed a decrease in galactosylation with high CDX1 mRNA expression. Previous reports showed that *CDX1, CDX2, HNF1A*, and *HNF1B* are involved in the transcriptional regulation of *B3GALT5*, the gene encoding for the enzyme responsible for type-1 chain (-Galβ1–3GlcNAcβ-) expression on glycolipids and glycoproteins, with preferences for *O*-glycans and glycosphingolipid-glycans [61]. *B3GALT5* was found to be down-regulated in colon cancer cell lines, but up-regulated upon CaCo2 cell differentiation [61]. However, our mass spectrometric approach does not allow distinction between type-1 and type-2 chains (-Galβ1–4GlcNAc-), but enzyme levels showed an up-regulation of type-1 chain glycosyltransferases *B3GALT*s as well as *B3GNT*s according to observations by Isshiki et al. [61], while type-2 chain enzymes were decreased with high CDX1 expressions. Furthermore, many glycosyltransferases involved in glycan elongations act on *N*-glycans, *O*-glycans, glycosphingolipid-glycans, and/or others and changes on different glycan classes can be differently regulated. Therefore, it is striking that the *N*-glycomic data obtained by mass spectrometry are well in accordance with the transcriptomic data, suggesting that observed changes in, for example, fucosylation and sialylation are mainly attributed to *N*-glycans or are globally altered, but further investigations on *O*-glycans and glycosphingolipid-glycans are needed.

Our results further revealed the presence of terminal HexNAc residues to be increased in CDX1^high^ expressing cells. Glycan motifs that may contribute to this increase include LacdiNAc structures as well as the Sda antigen; the latter was shown to be expressed in normal colon tissues and decreased during colon cancer progression [62]. LacdiNAc, on the other hand, was associated with differentiation of mammary epithelial cells and tumor suppression in neuroblastoma, whereas its expression was increased in human prostate, ovarian, and pancreatic cancers [63]. Furthermore, bisecting GlcNAc containing glycans contribute to this group and corresponding *MGAT3* gene expression was up-regulated with CDX1^high^ cells. Several reports describe reduced bisection in several tumors [64] and it was shown that it suppresses *N*-glycan branching by *MGAT5* [65], as well as α2,3-sialyation [66], the latter being reduced in CDX1^high^ cells. 

Strikingly, observed glycan phenotypes for CDX1^low^ cells matched largely those described for cells undergoing EMT, which involves the loss of epithelial markers such as E-cadherin and gain of mesenchymal markers such as vimentin. EMT-associated glycan changes have previously been described and include the aforementioned loss of antenna-fucosylation, but also decrease in antennarity of *N*-glycans and bisection, whereas enhanced levels of high-mannose type glycans, core-fucosylation, and corresponding *FUT8* expression, as well as increased α2,6-sialylation and *ST6GAL1* gene expression, were observed [67,68]. Inhibition of *FUT3* and *FUT6* has been shown to affect TGF-β receptor glycosylation, resulting in decreased fucosylation as well as *FUT3/6*-associated sialyl Lewis antigens and altered TGF-β-mediated EMT and invasion in colorectal cancer cells [69]. Furthermore, loss of CDX 1 and/or CDX2 was shown to impact TGF beta signaling and tumor invasion in murine APC mutant colon cancer models. Upon loss of CDX1/2, cells were poorly differentiated, invasive, and developed a villous morphology, which was accompanied by the loss of the epithelial marker E-cadherin, whereas expression of vimentin, *Twist1, Zeb1,* and *Zeb2* was induced [70]. Finally, *HNF1A* and *HNF4A* have been described to prevent EMT in liver cancer [71,72,73]. The role of glycosylation in EMT and a potential involvement of CDX1 clearly needs further investigation.

While the involvement of transcription factors was shown for fucosyltransferases, reports on sialyltransferases and galactosyltransferases involved in the elongation of *N*-glycans are still lacking and more research on the integrated regulation, as well as competition of glycosyltransferases, is needed. Also, the role of LacdiNAc structures and other terminal HexNAc epitopes in differentiation and colorectal cancer needs further investigation.

## 5. Conclusions

Our data, in combination with reports from literature, suggest that CDX1 (and CDX2) are involved in the regulation of multiple glycosyltransferases, especially fucosyltransferases, likely via interactions with other transcription factors, such as *HNF4A* and *HNF1A*. However, CDX genes may (additionally) influence the expression of fucose-carrying glycoproteins themselves, thereby leading to an *N*-glycan phenotype with enhanced multi-fucosylation. Taking into account the interaction of the two CDX-proteins with *HNF1A, HNF4A*, and *HNF1B* and together with the proven role of *HNF1A* and *HNF4A* in the regulation of fucosylation, we hypothesize a cooperation of these transcription factors being involved in the expression of *FUT* genes and thereby increasing fucosylation of glycoproteins with high CDX1/CDX2 expression (summarized in Figure 6). Certainly, more mechanistic studies are needed to elucidate the role of CDX1 in glycosyltransferase regulations.

## Figures and Tables

**Figure 1 cells-08-00273-f001:**
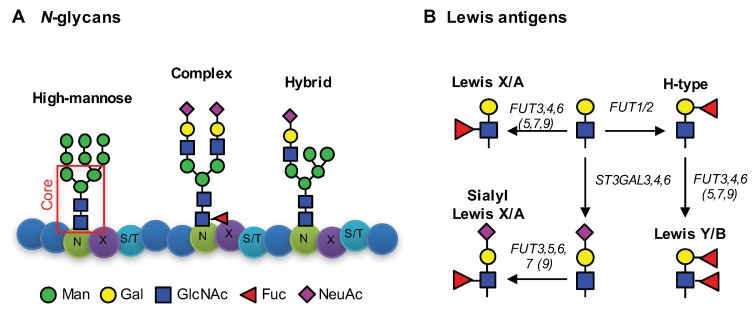
*N*-glycans and Lewis antigens. (**A**) *N*-glycans are attached to an asparagine (N) in the consensus sequence N-X-S/T with X any amino acid except proline, followed by serine (S) or threonine (T). Main monosaccharides involved in *N*-glycosylation are mannoses (Man), galactoses (Gal), *N*-acetylglucosamine (GlcNAc), fucoses (Fuc), and *N*-aceylneuraminic acid (NeuAc). *N*-glycans share a common core-structure, consisting of two GlcNAc and three Man. Depending on the elongation, three *N*-glycan types are differentiated, as follows: i) High-mannose type *N*-glycans, ii) complex-type *N*-glycans, and iii) hybrid-type *N*-glycans. The illustrated glycans represent examples. The number of monosaccharides added can vary and complex type glycans can exhibit more than two antennae. A detailed description of *N*-glycosylation is given by Stanley [4]. (**B**) Depiction of different Lewis-antigens and involved glycosyltransferase genes. Fucosyltransferase genes *FUT3,4,5,6,7,9* are involved in fucosylation α1,3- and α1,4-fucosylation of antennae GlcNAc, while *FUT1,2* attach α1,2-fucosylation to Gal-residues. Activity of several α2,3-sialyltransferase (*ST3GAL* genes) attach NeuAc residues to galactoses to form sialyl Lewis antigens.

**Figure 2 cells-08-00273-f002:**
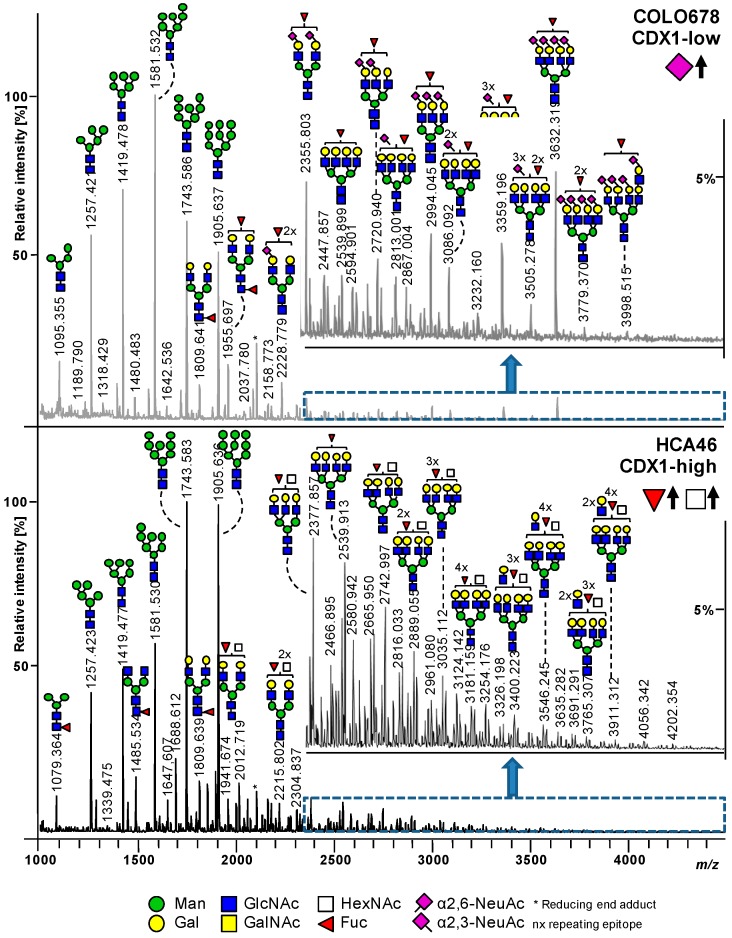
Exemplary MALDI-TOF-MS spectra. MALDI-TOF-MS spectrum of CDX1-low expressing cell line COLO678 (upper spectrum), and of CDX1-high expressing cell line HCA46 (lower spectrum). Spectra were recorded in positive ion reflectron mode on a Bruker UltrafleXtreme mass spectrometer. On the y-axis, the relative intensity is given with 100% corresponding to the highest peak in each spectrum. The spectrum range of *m/z* 2350 to *m/z* 4600 is enlarged in the inset. Main peaks are annotated with glycan cartoons, representing compositions, and the presence of additional structural isomers cannot be excluded. To simplify the cartoons, repeating epitopes are indicated as “nx”. Green circle = mannose, Man; yellow circle = galactose, Gal; blue square = *N*-acetylglucosamine, GlcNAc; white square = *N*-acetylhexosamine, HexNAc; red triangle = fucose, Fuc; purple diamond = sialic acid, *N*-acetylneuraminic acid, NeuAc. Differences in *N*-acetylneuraminic acid linkages are indicated using different angles.

**Figure 3 cells-08-00273-f003:**
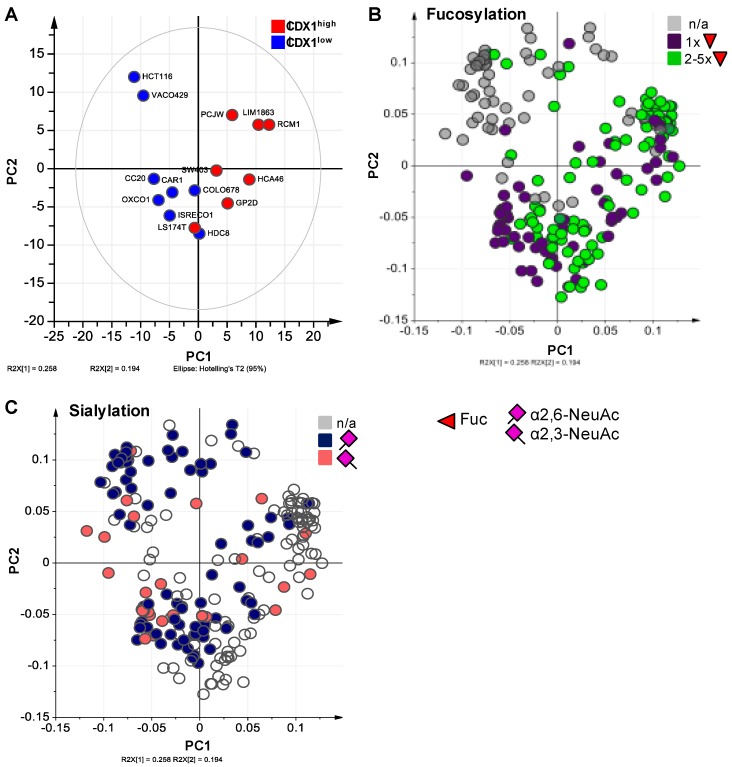
Principle component analysis of *N*-glycomic profiles. Principle component analysis (PCA) was performed to explore differences in *N*-glycosylation between CDX1-high and CDX1-low expressing cells. (**A**) Score plot of principal components (PC) 1 vs. 2 explaining 25.8% and 19.4% of the data, respectively. (**B**) Corresponding loading plot with color indications for multi- (green) and mono-fucosylated (purple) individual *N*-glycans, and (**C**) the same loading plot with color indications for α2,3-sialylated (rose) and α2,6-sialylated (blue) *N*-glycans. Red triangle = fucose, Fuc; purple diamond = sialic acid, *N*-acetylneuraminic acid, NeuAc. Differences in *N*-acetylneuraminic acid linkages are indicated using different angles.

**Figure 4 cells-08-00273-f004:**
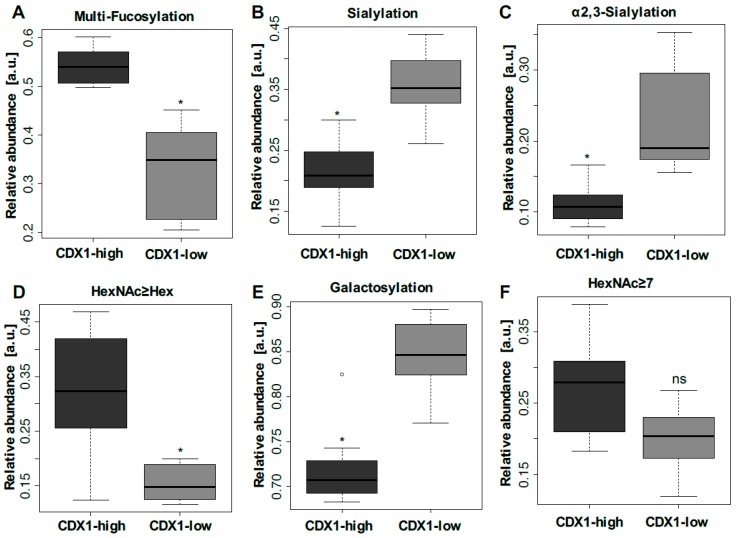
Differentially expressed *N*-glycan traits. Derived *N*-glycan traits were calculated to evaluate glycosylation characteristics associated with CDX1 mRNA expression. Differential expressions were observed for (**A**) multi-fucosylation (more than one fucose) indicative for antenna fucosylation, (**B**) overall sialylation (**C**), α2,3-sialylation, (**D**) *N*-glycans with more or equal *N*-acetylhexosamines than hexoses (HexNAc ≥ Hex), (**E**) galactosylation per antenna, and (**F**) *N*-glycans with seven or more HexNAcs. Differences were evaluated by Mann–Whitney test for derived traits and significances (*p*-value < 0.05) after multiple testing corrections are indicated (*), ns = non-significant. Boxplots consistently indicate the median and interquartile range.

**Figure 5 cells-08-00273-f005:**
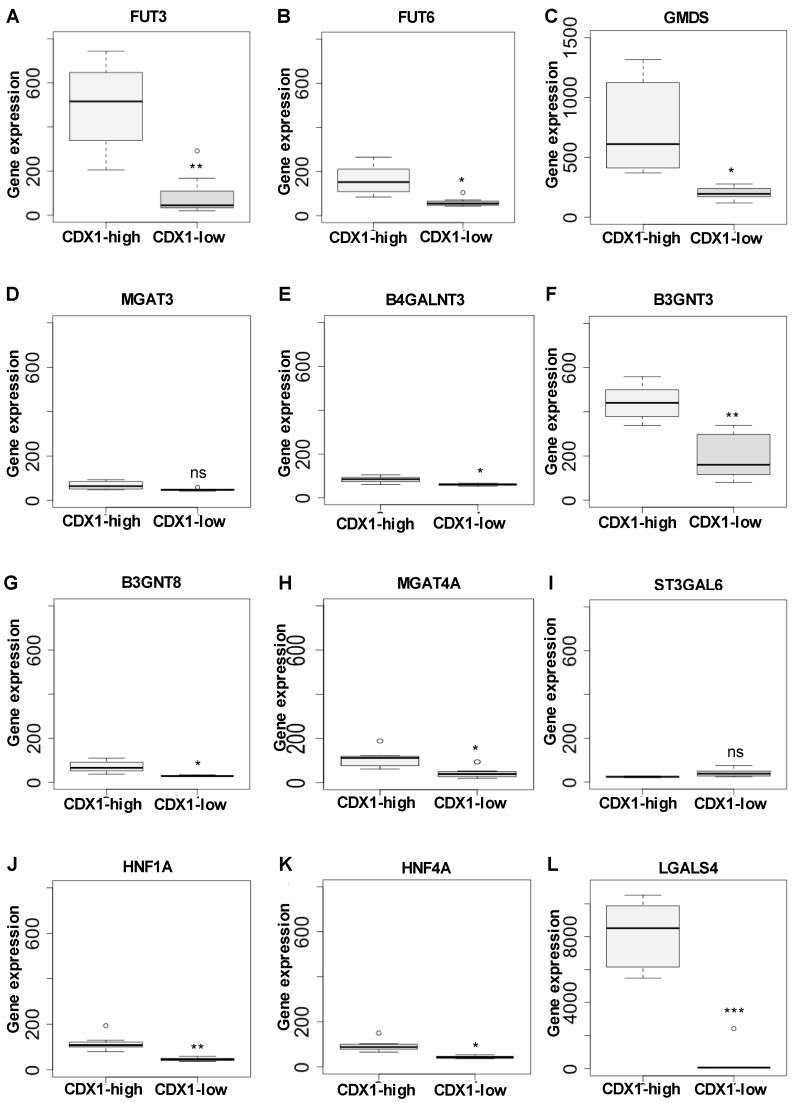
Differentially expressed genes related to glycosylation. *N*-glycomic phenotypes of the eight CDX1-high vs. eight CDX1-low cell lines were further compared to transcriptomic data of the same cell lines obtained from a gene expression microarray analysis using the Human genome U133+2 Affimetrix chips. The Y axis shows the linear normalized fluorescent intensities of the respective probesets. Differential expressions were observed for fucosyltransferase genes (**A**) *FUT3*, (**B**) *FUT6*, and (**C**) *GMDS*, for (**D**) MGAT3 (bisecting GlcNAc) and (**E**) B4GALNT3 (LacdiNAc), as well as (**F**) B3GNT3 (poly-LacNAc), (**G**) B3GNT8 (poly-LacNAc), (**H**) MGAT4A (α1-3-branching), and (**I**) α2,3-sialyltransferase gene ST3GAL6. Furthermore, transcription factors (**J**) hepatocyte nuclear factor (HNF)1A, and (**K**) HNF4A, as well as (**L**) soluble galectin 4 (LGALS4) showed increased expression with high CDX1 expression in the set of 16 colorectal cancer cell lines. Differences were evaluated by a *t-test*. Boxplots indicate the median and interquartile range. Significances after multiple testing correction are indicated and correspond to * *p*-value < 0.05, ** *p*-value < 0.01, *** *p*-value < 0.001; ns non-significant.

**Figure 6 cells-08-00273-f006:**
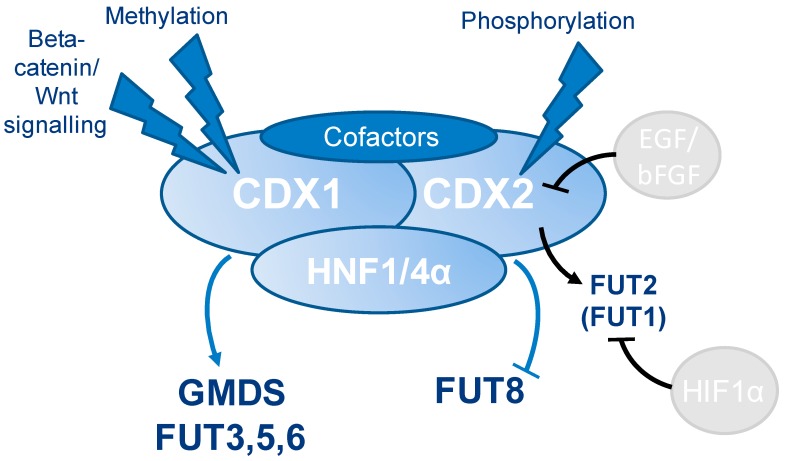
Model for regulation of fucosylation. Based on our findings as well as reports from literature (Lauc, G. et al. (2010) PLoS Genet **6**; Guo, R. J., Suh, E. R. & Lynch, J. P. (2004) Cancer Biol Ther **3**, 593-601) we hypothesize that antenna-fucosylation is regulated by an interplay of several transcription factors, especially CDX1, CDX2, HNF1A, and HNF4A, which can be affected by several signaling pathways, promotor methylation, as well as treatment with epidermal growth factor (EGF)/bFGF (Sakuma, K., Aoki, M. & Kannagi, R. (2012) Proceedings of the National Academy of Sciences of the United States of America **109**, 7776-7781) or under hypoxia (Belo, A. I. et al. (2015) FEBS Lett **589**, 2359-2366).

**Table 1 cells-08-00273-t001:** **Cell line characteristics.** CDX1 mRNA expression levels are given and cell lines below expression levels of 60 units were considered CDX1 low expressing cells; a.u., arbitrary units; n.a., data not available.

Cell Lines	CDX1 mRNA [a.u.]	Tumor	Duke’s/grade	Differentiation	Lumen Formation
HCA46	3308.59	sigmoid colon adenocarcinoma	C	poorly	Lumen
HCC56	4115.35	colon adenocarcinoma/liver?	n.a.	moderately	Dense
GP2D	3058.48	colon adenocarcinoma	B	poorly	Intermediate
PCJW	2598.75	colon adenocarcinoma	C	poorly	Intermediate
LS174T	1147.36	colon adenocarcinoma	B/grade I	well	Intermediate
LIM1863	2862.38	colon adenocarcinoma	C/grade III	poorly	Lumen
SW403	2964.14	colon adenocarcinoma	C/grade III	n.a.	Intermediate
RCM1	1152.22	colon adenocarcinoma	n.a.	n.a.	Intermediate
ISRECO1	32.10	colon adenocarcinoma	n.a.	n.a.	n.a.
VACO429	28.69	colon adenocarcinoma	n.a.	n.a.	Dense
HCT116	59.94	colon adenocarcinoma	grade IV	poorly	Dense
CC20	59.51	sigmoid colon adenocarcinoma	B	well	Network/stellate
CAR1	49.35	colon adenocarcinoma	n.a.	n.a.	Dense
COLO678	16.21	colon adenocarcinoma, metastatic lymph node	n.a.	n.a.	Dense
HDC8	29.31	colon adenocarcinoma	C/grade III	n.a.	Intermediate
OXCO1	49.69	colon adenocarcinoma	n.a.	n.a.	n.a.

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
