# Peer review of "N-Glycomic and Transcriptomic Changes Associated with CDX1 mRNA Expression in Colorectal Cancer Cell Lines"

_cells, 2019, doi:10.3390/cells8030273_

Round 1
Reviewer 1 Report
This paper addresses the relationship of N-glycosylation as a posttranslational modification to the caudal-related homeobox protein 1 (CDX1) mRNA expression in colorectal cancer cell lines. The following points should be addressed:
1. Is there any evidence for sulphation of the CDX1 N-glycans? This form of N-glycan modification has been reported, e.g. (Roux, Holojda et al. 1988, Baenziger 1995, Galustian, Lawson et al. 1997, Yoshimura, Hayashi et al. 2017).
2. A short comment should be added to the text to draw attention to the importance of FUT2 in the formation of blood group antigens and secretor status, see e.g. (Mollicone, Cailleau et al. 1995, Le Pendu 2004, Padro, Mejias-Luque et al. 2011) and whether this is relevant to CDX1 and CDX2 N-glycosylation.
3. The data for sialic acid expression in N-glycans does not mention the a2-8/9 linkage for disialyl or polysialyl linkages, generated by the ST8sia sialyltransferases e.g. (Chang, Mir et al. 2009), which has been reported for glycoprotein N-glycans e.g. (Finne, Finne et al. 1983, Galuska, Geyer et al. 2007) and for a homeobox peptide (Joliot, Le Roux et al. 1993, Joliot, Le Roux et al. 1994). The analysis reported in reference 35 (Reiding et al., Analytical Chemistry, 2014), use sialyl lactose to discriminate a2-3 and a2-6 linkages. Assessment of a2-8 linkages could be made using disialyl-lactose, present in bovine colostrum and available from Carbosynth, Newbury, UK. A comment should be added to the text to draw attention to this feature.
4. Following on from point 3 above, the sialic acids are known to occur as O-acetylated forms, which have biological significance, see e.g. (Diaz, Higa et al. 1989, Klein, Krishna et al. 1994, Mann, Klussmann et al. 1997, Mandal, Mandal et al. 2012, Langereis, Bakkers et al. 2015). Are the sialic acids on the CDX1 N-glycans O-acetylated? Could this have a biological significance?
5. The paper identifies GMDS as an enzyme on the L-Fucose metabolic pathway that causes abberations if deleted or inactive. There are other , non-fucosyltransferase proteins associated with L-Fucose metabolism which could also play a role in the regulation of L-Fucose expression, including the GDP-Fucose transporter into the ER/Golgi, to enable normal fucosylation see e.g. (Freeze, Hart et al. 2017). Is there any evidence for the involvement of this transporter in the regulation of CDX1 fucosylation?
6. Figure 4 is wrongly labeled and the legend is also incorrect. As a result the cross-references to this Figure in the text are also wrong. The Figure, Legend and normal text should be reviewed to correct these errors.
7. A number of the genes shown in Figure 5 show low expression levels for both CDX1high and CDX1low, this applies to MGAT3/B4GALNT3/B3GNT8/MGAT4A/ST3GAL6/HNF1A and HNF4A. Does the level of expression play a role in regulation?
8. The data shown in Supplementary Table S5A show significant p values for B3GNT3 and B3GNT8. However, there is no discussion of the relevance of these results in the paper. A comment should be added to the paper.
MINOR POINTS
9. In Figure 3, Fucose is shown as a red triangle and mentioned in the legend, but does not appear anywhere in Figure 3. Please check whether this is necessary for this Figure.
10. A number of the References are incomplete. This applies to 1, 6, 10, 33, 49 and 59. The reference list should be checked for completion throughout.
Literature cited in the report
Baenziger, J. U. (1995). "Regarding the glycoprotein hormones and their sulphated oligosaccharides." Glycobiology 5(5): 459.
Chang, L. Y., A. M. Mir, C. Thisse, Y. Guerardel, P. Delannoy, B. Thisse and A. Harduin-Lepers (2009). "Molecular cloning and characterization of the expression pattern of the zebrafish alpha2, 8-sialyltransferases (ST8Sia) in the developing nervous system." Glycoconj J 26(3): 263-275.
Diaz, S., H. H. Higa, B. K. Hayes and A. Varki (1989). "O-acetylation and de-O-acetylation of sialic acids. 7- and 9-o-acetylation of alpha 2,6-linked sialic acids on endogenous N-linked glycans in rat liver Golgi vesicles." J Biol Chem 264(32): 19416-19426.
Finne, J., U. Finne, H. Deagostini-Bazin and C. Goridis (1983). "Occurrence of alpha 2-8 linked polysialosyl units in a neural cell adhesion molecule." Biochem Biophys Res Commun 112(2): 482-487.
Freeze, H. H., G. W. Hart and R. L. Schnaar (2017). Glycosylation Precursors. Essentials of Glycobiology. A. Varki, R. D. Cummings, J. D. Esko et al. New York, Cold Spring Harbor Laboratory Press: 51-63.
Galuska, S. P., R. Geyer, M. Muhlenhoff and H. Geyer (2007). "Characterization of oligo- and polysialic acids by MALDI-TOF-MS." Anal Chem 79(18): 7161-7169.
Galustian, C., A. M. Lawson, S. Komba, H. Ishida, M. Kiso and T. Feizi (1997). "Sialyl-Lewis(x) sequence 6-O-sulfated at N-acetylglucosamine rather than at galactose is the preferred ligand for L-selectin and de-N-acetylation of the sialic acid enhances the binding strength." Biochem. Biophys. Res. Commun. 240: 748-751.
Joliot, A., I. Le Roux, M. Volovitch, E. Bloch-Gallego and A. Prochiantz (1993). "Neurotrophic activity of an homeobox peptide." Ann Genet 36(1): 70-72.
Joliot, A., I. Le Roux, M. Volovitch, E. Bloch-Gallego and A. Prochiantz (1994). "Neurotrophic activity of a homeobox peptide." Prog Neurobiol 42(2): 309-311.
Klein, A., M. Krishna, N. M. Varki and A. Varki (1994). "9-O-acetylated sialic acids have widespread but selective expression: Analysis using a chimeric dual-function probe derived from influenza C hemagglutinin-esterase." Proc. Natl. Acad, Sci. U.S.A. 91: 7782-7786.
Langereis, M. A., M. J. Bakkers, L. Deng, V. Padler-Karavani, S. J. Vervoort, R. J. Hulswit, A. L. van Vliet, G. J. Gerwig, S. A. de Poot, W. Boot, A. M. van Ederen, B. A. Heesters, C. M. van der Loos, F. J. van Kuppeveld, H. Yu, E. G. Huizinga, X. Chen, A. Varki, J. P. Kamerling and R. J. de Groot (2015). "Complexity and Diversity of the Mammalian Sialome Revealed by Nidovirus Virolectins." Cell Rep 11(12): 1966-1978.
Le Pendu, J. (2004). "Histo-blood group antigen and human milk oligosaccharides: genetic polymorphism and risk of infectious diseases." Adv Exp Med Biol 554: 135-143.
Mandal, C., C. Mandal, S. Chandra, R. Schauer and C. Mandal (2012). "Regulation of O-acetylation of sialic acids by sialate-O-acetyltransferase and sialate-O-acetylesterase activities in childhood acute lymphoblastic leukemia." Glycobiology 22(1): 70-83.
Mann, B., E. Klussmann, V. Vandamme-Feldhaus, M. Iwersen, M. L. Hanski, E. O. Riecken, H. J. Buhr, R. Schauer, Y. S. Kim and C. Hanski (1997). "Low O-acetylation of sialyl-Le(x) contributes to its overexpression in colon carcinoma metastases." Int J Cancer 72(2): 258-264.
Mollicone, R., A. Cailleau and R. Oriol (1995). "Molecular genetics of H, Se, Lewis and other fucosyltransferase genes." Transfus Clin Biol 2(4): 235-242.
Padro, M., R. Mejias-Luque, L. Cobler, M. Garrido, M. Perez-Garay, S. Puig, R. Peracaula and C. de Bolos (2011). "Regulation of glycosyltransferases and Lewis antigens expression by IL-1beta and IL-6 in human gastric cancer cells." Glycoconj J 28(2): 99-110.
Roux, L., S. Holojda, G. Sundblad, H. H. Freeze and A. Varki (1988). "Sulfated N-linked oligosaccharides in mammalian cells. I. Complex-type chains with sialic acids and O-sulfate esters." J Biol Chem 263(18): 8879-8889.
Yoshimura, T., A. Hayashi, M. Handa-Narumi, H. Yagi, N. Ohno, T. Koike, Y. Yamaguchi, K. Uchimura, K. Kadomatsu, J. Sedzik, K. Kitamura, K. Kato, B. D. Trapp, H. Baba and K. Ikenaka (2017). "GlcNAc6ST-1 regulates sulfation of N-glycans and myelination in the peripheral nervous system." Sci Rep 7: 42257.
Author Response
Dear reviewers,
We would like to thank you very much for handling our manuscript. We also thank the reviewers for their comments and comprehensive advice on our manuscript. We revised the manuscript in accordance with reviewers’ suggestions, which are critically important to our work. Please find below our responses.
We hope that you will find the revised manuscript with the title “N-glycomic and transcriptomic changes associated with CDX1 mRNA expression in colorectal cancer cell lines” suited for publication in the Cells’ special issue Functional Glycomics.
Yours sincerely,
Dr. Stephanie Holst and Prof. Manfred Wuhrer
REVIEWER 1
This paper addresses the relationship of N-glycosylation as a posttranslational modification to the caudal-related homeobox protein 1 (CDX1) mRNA expression in colorectal cancer cell lines. The following points should be addressed:
1. Is there any evidence for sulphation of the CDX1 N-glycans? This form of N-glycan modification has been reported, e.g. (Roux, Holojda et al. 1988, Baenziger 1995, Galustian, Lawson et al. 1997, Yoshimura, Hayashi et al. 2017).
In this manuscript, our aim was to investigate whether CDX1 expression in colorectal cancer cell lines associates with their N-glycosylation. Therefore, N-glycosylation analysis has been performed on cell lysates. We have not analyzed the glycosylation of CDX1 protein itself which would require an immunoprecipitation step. Further, to our knowledge, no N-glycosylation of the CDX1 protein has been reported yet which is expected considering that CDX1 is an intracellular protein. Regarding the sulfation of N-glycans, we previously reported an increase of sulfated N-glycans in colon tumor tissues as compared to control tissues [1]. In the current study, N-glycans were analyzed by MALDI-TOF-MS in positive ion mode which is not suited to detect sulfation modification (negatively charged) on N-glycans derived from colorectal cancer cell lines. Nonetheless, carbohydrate sulfotrasferases (CHST) which were included in the gene microarray did not show significant differences in expression between the here studied CDX1 high and CDX1 low expressing colorectal cell lines and alternative detection techniques were therefore not pursued.
2. A short comment should be added to the text to draw attention to the importance of FUT2 in the formation of blood group antigens and secretor status, see e.g. (Mollicone, Cailleau et al. 1995, Le Pendu 2004, Padro, Mejias-Luque et al. 2011) and whether this is relevant to CDX1 and CDX2 N-glycosylation.
We modified the text according to reviewer’s recommendations and added the following to the introduction:
‘While FUT3,4,5,6,7,9catalyze the addition of a fucose to the N-acetylglucosamine (GlcNAc) of the antenna in α1,3- and/or α1,4-linkage, FUT1and FUT2 are responsible for the addition of a fucose to the galactose (Gal) in α1,2-linkage, forming the H-type epitope. FUT2is further called the secretor gene and polymorphisms leading to an inactive FUT2lead to the absence of blood type epitopes in saliva and various epithelial cell types, the so-called non-secretor phenotype [2-3].’
In our study, both groups of cells express FUT2. Reports on the non-secretor phenotype have shown that polymorphisms of FUT2 lead to an inactive form, therefore causing this specific phenotype [2-3]. If FUT2 is transcriptionally down-regulated (by CDX1, CDX2 or other transcription factors) or influenced via cytokines as shown for IL-1β and IL-6 [4], this could also lead to variations of the secretor epitopes. However, to our knowledge CDX1 or CDX2 have not yet been associated with the secretor status. Furthermore, the expression of glycoproteins or glycolipids carrying histo blood group epitopes such as mucins can influence the abundance of these epitopes. In that context, CDX1 has been shown to directly regulate MUC2 [5].
3. The data for sialic acid expression in N-glycans does not mention the a2-8/9 linkage for disialyl or polysialyl linkages, generated by the ST8sia sialyltransferases e.g. (Chang, Mir et al. 2009), which has been reported for glycoprotein N-glycans e.g. (Finne, Finne et al. 1983, Galuska, Geyer et al. 2007) and for a homeobox peptide (Joliot, Le Roux et al. 1993, Joliot, Le Roux et al. 1994). The analysis reported in reference 35 (Reiding et al., Analytical Chemistry, 2014), use sialyl lactose to discriminate a2-3 and a2-6 linkages. Assessment of a2-8 linkages could be made using disialyl-lactose, present in bovine colostrum and available from Carbosynth, Newbury, UK. A comment should be added to the text to draw attention to this feature.
The reviewer is right: similar to α2,3-linked sialic acid, α2,8-linked sialic acid will be lactonized under the conditions of the derivatization step [6]. We therefore cannot differentiate between α2,3-linked sialic acid and α2,8-linked sialic on the basis of the observed masses alone. However, we previously performed in-depth MS/MS characterization of N-glycans from colorectal cancer cells and could not find evidence for a fragment corresponding to an antenna with two or more sialic acids [7]. Although N-glycans with terminal α2,8-linked sialic acids are expressed in cancer cells, these structures might be expressed at low levels as compared to N-glycans with terminal α2,6- and α2,3-linked sialic acids. Moreover, gene expression of ST8SIA1-5 was further not significantly different between the here studied CDX1 high and CDX1 low expressing colorectal cell lines. We added this paragraph as well in the discussion.
4. Following on from point 3 above, the sialic acids are known to occur as O-acetylated forms, which have biological significance, see e.g. (Diaz, Higa et al. 1989, Klein, Krishna et al. 1994, Mann, Klussmann et al. 1997, Mandal, Mandal et al. 2012, Langereis, Bakkers et al. 2015). Are the sialic acids on the CDX1 N-glycans O-acetylated? Could this have a biological significance?
Indeed, sialic acids can also be modified by O-acetyl-groups modulating the ligand function. O-actelyated glycans have been reported to be mainly present in the lower part of the intestinal tract [8]. Further, a decrease of O-acetylation has been observed with colon cancer progression [9]. In line, we previously detected O-acetylated sialic acids of glycosphingolipid glycans which were decreased in colorectal cancer tissues as compared to control tissues [10], but not on N-glycans of the same tissues [1]. The derivatization method applied in this study to characterize the N-glycans of colorectal cancer cell lines has been shown to preserve O-acetylation [11]. However, we could not find evidence of O-acetylated N-glycans in the investigated cell lines (CDX1 protein glycosylation has not been analyzed). We have added this paragraph as well in the discussion.
5. The paper identifies GMDS as an enzyme on the L-Fucose metabolic pathway that causes aberrations if deleted or inactive. There are other, non-fucosyltransferase proteins associated with L-Fucose metabolism which could also play a role in the regulation of L-Fucose expression, including the GDP-Fucose transporter into the ER/Golgi, to enable normal fucosylation see e.g. (Freeze, Hart et al. 2017). Is there any evidence for the involvement of this transporter in the regulation of CDX1 fucosylation?
To our knowledge, GMDS is the most studied gene with relation to aberrations in the fucose biosynthesis pathway. In our set of genes, we further saw that SLC35C1, a fucose transporter, was differentially expressed between the two studied groups of colorectal cancer cell lines. However, we decided to limit the results in the manuscript to glycosyltransferase genes or related genes which have known relevance; also, since the Affymetrix Human genome U133+2 chip used in this study does not contain the full spectrum of genes related to the fucose metabolism. We therefore cannot exclude the involvement of other genes.
6. Figure 4 is wrongly labeled and the legend is also incorrect. As a result, the cross-references to this Figure in the text are also wrong. The Figure, Legend and normal text should be reviewed to correct these errors.
We thank the reviewer for pointing out this mistake. We have corrected the figure which now matches the figure legend and the references in the text. Our apologies for this mistake.
7. A number of the genes shown in Figure 5 show low expression levels for both CDX1high and CDX1low, this applies to MGAT3/B4GALNT3/B3GNT8/MGAT4A/ST3GAL6/HNF1A and HNF4A. Does the level of expression play a role in regulation?
The gene expression data has been RMA-normalized (with GC correction) and does not display absolute values. Further, the gene expression level does not necessarily mean a low enzyme activity in case of the glycosyltransferases.
8. The data shown in Supplementary Table S5A show significant p values for B3GNT3 and B3GNT8. However, there is no discussion of the relevance of these results in the paper. A comment should be added to the paper.
According to the reviewer’s suggestion we have added the results in the manuscript:
3.5 CDX1 expression in CRC cell lines associated with higher branched N-glycan-derived traits
Finally,N-glycan structures with seven or more HexNAcs, indicative for branched structures or (poly) LacNAc repeats (–Galβ1–4GlcNAcβ1–3Galβ1–3/4–GlcNAcβ1–), showed a trend towards higher expression with high CDX1 expression in our previous data and were significantly higher expressed in CDX1highcell lines of the new set of cells as compared to CDX1lowexpressing cell lines with Æ27% vs. Æ20% (Fig. 4F, Supplementary Table S3). Since our MS data could not sufficiently differentiate between LacNAc-repeat and additional antenna, we next analyzed the glycosyltransferase expression data to see if this would give a more detailed insight. Genes encoding for beta-1,3-N-acetylglucosaminyltransferase 3 (B3GNT3) and B3GNT8, both involved in the synthesis of type-1 chains and poly-LacNAc repeats, were around 2-fold up-regulated with high CDX1 expression (Fig. 5F+G, Supplementary Table S5A) and B3GNT3 gene expression showed significant correlation with the MS trait HexNAc≥7 (Supplementary Table S6), suggesting the presence of LacNAc-repeat structures to a certain degree. Also, the glycosyltransferase encoded by gene MGAT4A, which is involved in the branching on the 1,3-arm of N-glycans to form tri- and tetra-antennary N-glycans, was correlated with the relative abundance of the MS N-glycan trait HexNAc≥7 (Supplementary Table S6). MGAT4A was further significantly higher expressed in CDX1highcells as compared to CDX1lowcells (Fig. 5H, Supplementary Table S5A). Of note, N-glycans featuring HexNAc≥Hex also contribute to this trait of N-glycan structures with seven or more HexNAcs, which is therefore not solely indicative for branching and poly-LacNAc repeats. In line, B3GNT3 and B3GNT8 also showed correlation with the MS trait HexNAc≥Hex (Supplementary Table S6).
MINOR POINTS
9. In Figure 3, Fucose is shown as a red triangle and mentioned in the legend, but does not appear anywhere in Figure 3. Please check whether this is necessary for this Figure.
Figure 3B shows the PCA score plot differentiating mono-fucosylation (one fucose, purple) and multi-fucosylation (two or more fucoses, green). The plot itself has a small legend which explains the colors for which we used the red triangle for the fucose.
10. A number of the References are incomplete. This applies to 1, 6, 10, 33, 49 and 59. The reference list should be checked for completion throughout.
We thank the reviewer for pointing out the incomplete references. We adjusted the references and complemented the missing information.
References used for the responses:
1. Balog, C. I. A.; Stavenhagen, K.; Fung, W. L. J.; Koeleman, C. A.; McDonnell, L. A.; Verhoeven, A.; Mesker, W. E.; Tollenaar, R. A. E. M.; Deelder, A. M.; Wuhrer, M., N-glycosylation of Colorectal Cancer Tissues: A Liquid Chromatography And Mass Spectrometry-Based Investigation. Mol. Cell. Proteomics 2012,11(9), 571–585.
2. Mollicone, R.; Cailleau, A.; Oriol, R., Molecular genetics of H, Se, Lewis and other fucosyltransferase genes. Transfus Clin Biol 1995,2(4), 235-42.
3. Le Pendu, J., Histo-blood group antigen and human milk oligosaccharides: genetic polymorphism and risk of infectious diseases. Advances in experimental medicine and biology 2004,554, 135-43.
4. Padro, M.; Mejias-Luque, R.; Cobler, L.; Garrido, M.; Perez-Garay, M.; Puig, S.; Peracaula, R.; de Bolos, C., Regulation of glycosyltransferases and Lewis antigens expression by IL-1beta and IL-6 in human gastric cancer cells. Glycoconjugate journal 2011,28(2), 99-110.
5. Mesquita, P.; Jonckheere, N.; Almeida, R.; Ducourouble, M. P.; Serpa, J.; Silva, E.; Pigny, P.; Silva, F. S.; Reis, C.; Silberg, D.; Van Seuningen, I.; David, L., Human MUC2 mucin gene is transcriptionally regulated by Cdx homeodomain proteins in gastrointestinal carcinoma cell lines. The Journal of biological chemistry 2003,278(51), 51549-56.
6. Hanamatsu, H.; Nishikaze, T.; Miura, N.; Piao, J.; Okada, K.; Sekiya, S.; Iwamoto, S.; Sakamoto, N.; Tanaka, K.; Furukawa, J. I., Sialic Acid Linkage Specific Derivatization of Glycosphingolipid Glycans by Ring-Opening Aminolysis of Lactones. Analytical chemistry 2018,90(22), 13193-13199.
7. Holst, S.; Deuss, A. J.; van Pelt, G. W.; van Vliet, S. J.; Garcia-Vallejo, J. J.; Koeleman, C. A.; Deelder, A. M.; Mesker, W. E.; Tollenaar, R. A.; Rombouts, Y.; Wuhrer, M., N-glycosylation Profiling of Colorectal Cancer Cell Lines Reveals Association of Fucosylation with Differentiation and Caudal Type Homebox 1 (CDX1)/Villin mRNA Expression. Molecular & cellular proteomics : MCP 2016,15(1), 124-40.
8. Varki, A., Sialic acids in human health and disease. Trends in molecular medicine 2008,14(8), 351-60.
9. Corfield, A. P.; Myerscough, N.; Warren, B. F.; Durdey, P.; Paraskeva, C.; Schauer, R., Reduction of sialic acid O-acetylation in human colonic mucins in the adenoma-carcinoma sequence. Glycoconjugate J. 1999,16(6), 307-317.
10. Holst, S.; Stavenhagen, K.; Balog, C. I.; Koeleman, C. A.; McDonnell, L. M.; Mayboroda, O. A.; Verhoeven, A.; Mesker, W. E.; Tollenaar, R. A.; Deelder, A. M.; Wuhrer, M., Investigations on aberrant glycosylation of glycosphingolipids in colorectal cancer tissues using liquid chromatography and matrix-assisted laser desorption time-of-flight mass spectrometry (MALDI-TOF-MS). Molecular & cellular proteomics : MCP 2013,12(11), 3081-93.
11. Falck, D.; Haberger, M.; Plomp, R.; Hook, M.; Bulau, P.; Wuhrer, M.; Reusch, D., Affinity purification of erythropoietin from cell culture supernatant combined with MALDI-TOF-MS analysis of erythropoietin N-glycosylation. Scientific reports 2017,7(1), 5324.
Reviewer 2 Report
The manuscript reports the N-glycomic and transcriptomic changes induced by the CDX1 expression in CRC cell lines.
The approach is good and the results are sound. The discussion and conclusions are clear and adequate.
Overall the mansucript is ready for publication.
Author Response
Dear reviewers,
We would like to thank you very much for handling our manuscript. We also thank the reviewers for their comments and comprehensive advice on our manuscript. We revised the manuscript in accordance with reviewers’ suggestions, which are critically important to our work. Please find below our responses.
We hope that you will find the revised manuscript with the title “N-glycomic and transcriptomic changes associated with CDX1 mRNA expression in colorectal cancer cell lines” suited for publication in the Cells’ special issue Functional Glycomics.
Yours sincerely,
Dr. Stephanie Holst and Prof. Manfred Wuhrer
REVIEWER 2
Comments and Suggestions for Authors
The manuscript reports the N-glycomic and transcriptomic changes induced by the CDX1 expression in CRC cell lines.
The approach is good and the results are sound. The discussion and conclusions are clear and adequate.
Overall the manuscript is ready for publication
We thank the reviewer for the comments and the support for publication of our manuscript.